# Synthesis, properties, and catalysis of p-block complexes supported by bis(arylimino) acenaphthene ligands

Jingyi Wang[1], Han Sen Soo [1✉] & Felipe Garcia [1✉]

Bis(arylimino)acenaphthene (Ar-BIAN) ligands have been recognized as robust scaffolds for metal complexes since the 1990 s and most of their coordination chemistry was developed with transition metals. Notably, there have been relatively few reports on complexes comprising main group elements, especially those capitalizing on the redox non-innocence of Ar-BIAN ligands supporting *p*-block elements. Here we present an overview of synthetic approaches to Ar-BIAN ligands and their p-block complexes using conventional solution-based methodologies and environmentally-benign mechanochemical routes. This is followed by a discussion on their catalytic properties, including comparisons to transition metal counterparts, as well as key structural and electronic properties of p-block Ar-BIAN complexes.

D uring the last decade, the development of metal complexes supported by redox-active ligands has attracted considerable interest in both academic and industrial settings, owing to their versatile electronic structures and potential applications in catalysis. The demonstrated ability of these ligands to display multiple consecutive oxidation states, together with their coordination to metal centers, induces both radical reactivity and electron-reservoir behavior, making them ideal ligands to support a wide range of catalytic transformations, examples of which are illustrated in Fig. 1.

Bis(imino)acenaphthene (BIAN) is a family of diimine ligands that can be considered as a fusion product between 1,4-diaza-1,3-butadiene (DAB) and a naphthalene unit with great potential as redox noninnocent ligands. Although they have been known since the 1960s, interest in the use of BIAN derivatives as ligands only intensified after the 1990s[1]. The BIAN compounds with aryl substituents on the diimine nitrogen atoms are named bis(arylimino)acenaphthene (Ar-BIAN), whereas those with alkyl substituents are bis(alkylimino)acenaphthene (R-BIAN). The two imine functionalities are often orthogonal to the naphthalene unit and therefore, the arylimines are usually not conjugated with the π-accepting framework. In contrast to the R-DAB ligands in which the imine nitrogen atoms preferentially adopt the *s-trans* conformation, the

[1] Division of Chemistry and Biological Chemistry, School of Physical and Mathematical Sciences, Nanyang Technological University, 21 Nanyang Link, 637371 Singapore, Singapore. ✉email: Hansen@ntu.edu.sg; fgarcia@ntu.edu.sg

**Fig. 1 Transformations mediated by metal complexes comprising redox noninnocent imino-type ligands.** Examples of key selected transformations: C–C bond formation[12], epoxidation reactions[51], C–H bond activation reactions[12], and hydrogenation reactions[12].

**Fig. 2 Different conformations and valence states of Ar-BIAN. a** Structural conformations of 1,4-diaza-1,3-butadiene (DAB) and bis(imino)acenaphthene (BIAN) compounds. **b** Representation of the formal charge-localized structures on Ar-BIAN upon reduction.

rigid naphthalene backbone in Ar-BIAN ligands prevents rotation around the diimine C–C bond, thus locking them in *s-cis* conformations that enable them to coordinate readily to metal centers (Fig. 2a)[2]. An essential feature of BIAN ligands is their ability to accept electrons. Their strong electron-accepting properties are evident from (i) the ability of the α-diimines to stabilize a range of photochemically[3–5] and electrochemically[6–8] generated open-

shell complexes through delocalization of the electron density into the antibonding orbitals, and (ii) the ready reduction of the naphthalene unit by alkali metals[9]. As illustrated in Fig. 2b, reduction of Ar-BIAN by a one-electron process first produces a radical anion that is delocalized over the NCCN fragment. This is followed by a second electron reduction, which results in the formation of a dianionic ene-diamide. The third and fourth

electron reductions will then be delocalized into the naphthalene backbone, which disrupts the aromaticity[1]. Accordingly, Fedushkin et al. had demonstrated that Ar-BIAN ligands can accept up to four electrons to form stable mono-, di-, tri-, and tetraanions by conducting a systematic synthetic study on the reduction of Dipp-BIAN (Dipp = 2,6-diisopropylphenyl) with sodium metal[10]. Owing to this facile electron-accepting property, Ar-BIAN ligands are widely recognized as redox noninnocent ligands able to support multiple formal metal redox states.

The stabilities of metal complexes formed with Ar-BIAN ligands can be controlled by the diimine substituents, with more electron-rich substituents likely to facilitate stronger binding to higher oxidation-state metal centers[11]. To evaluate the coordination abilities, Gasperini et al. compared the binding constants among a series of Ar-BIAN complexes with palladium(0) and (II), and proposed that the chelating strengths of the bidentate Ar-BIAN ligands fall between the acyclic Ph-DAB ligands (Ph-DAB = diphenyldiazabutadiene) and the more popular π-accepting ligands, such as phenanthroline and bipyridine[11].

To date, transition-metal BIAN complexes have been extensively studied and applied as catalysts for important chemical transformations[12–16], pioneered by the coordination chemistry of late-transition metals by the Benedix[2] and Templeton groups[17]. In contrast, there have been few reports devoted to the chemistry of BIAN ligands with the main group, especially p-block elements, despite their tantalizing prospects as transition-metal-like, redox-active systems. In this review, we first examine selected examples of the latest breakthroughs, since the authoritative work by Cowley et al. in 2009[18], in the chemistry of Ar-BIAN ligands with main-group elements. Subsequently, we summarize the field on the solid-state syntheses of main-group Ar-BIAN complexes. Finally, we highlight the photophysical and electrochemical properties of main-group Ar-BIAN complexes, and the potential implications for catalysis. Together with the emergence of solid-state mechanochemical syntheses of these systems, we envision bright prospects in the adoption of Ar-BIAN main-group complexes for catalysis and optoelectronic applications that exploit their redox versatility.

**Synthesis of Ar-BIAN ligands**. *Traditional solution-based synthesis of Ar-BIAN ligands and complexes*: Ar-BIAN ligands have typically been synthesized via condensation reactions between acenaphthoquinone and the corresponding aniline under acidic conditions. For example, Dipp-BIAN was synthesized by heating acenaphthoquinone with 2,6-diisopropylaniline in acetic acid for 1 h at reflux[2], while o-CF3Ar-BIAN (o-CF3 = o-tri-fluoromethyl) was formed by refluxing acenaphthoquinone with o-trifluoromethylaniline in a mixture of toluene and sulfuric acid for 3 days using a Dean–Stark trap to isolate the water that resulted[19]. In many other cases, templating with either zinc chloride (ZnCl$_2$) or nickel bromide (NiBr$_2$) was necessary before removal of the metal ion to furnish the desired ligand (Fig. 3a)[2]. Accordingly, a number of Ar-BIAN ligands, such as p-MeOAr-BIAN (p-methoxyphenyl), p-NMe2Ar-BIAN (p-dimethylamino-phenyl), p-MeAr-BIAN (p-methylphenyl), p-BrAr-BIAN (p-bro-mophenyl), and p-ClAr-BIAN (p-chlorophenyl) were synthesized by refluxing acenaphthoquinone with two equivalents or small excesses of the corresponding aniline in acetic acid in the presence of excess ZnCl$_2$ or NiBr$_2$, followed by demetallation using potassium carbonate (K$_2$CO$_3$) or sodium oxalate (Na$_2$C$_2$O$_4$)[20]. Besides serving as a template around which the Ar-BIAN is formed[21,22], Ragaini et al. suggested that another driving force for the condensation with metal halides is the precipitation of the resulting metal Ar-BIAN complexes due to their lower solubilities in the reaction media compared with the starting materials[21].

Beyond the simple, symmetric Ar-BIAN ligands, the synthesis of asymmetric variants from two different anilines had been reported by Ragaini et al. in 2004[23]. Two strategies were described, the first involving a transimination starting from a symmetric Ar-BIAN Zn complex bearing electron-withdrawing CF$_3$ groups on the diimine moiety[23]. In the second approach, a two-step process starting by condensation of acenaphthenequinone with 3,5-bis(trifluoromethyl)aniline to form the monosubstituted intermediate, was followed by a second ZnCl$_2$-templated condensation with an aniline-possessing electron-donating group (s) (Fig. 3b)[23]. These synthetic procedures rely on the large electronic differences and hence disparities in the kinetics of condensation between the two anilines with the acenaphthene-quinone. Even more sophisticated tridentate ligands based on BIAN with additional pendant O, P, and S donor atoms were also synthesized through a stepwise procedure that involved initial formation of a monoimine using a bulky aniline, followed by reaction of the intermediate with another amine tethered to the remaining donor group (Fig. 3c)[24].

*Mechanochemical synthesis of Ar-BIAN*: In recent years, mechanochemical synthesis has been gaining momentum for the preparation of a variety of organic, organometallic, polymeric, nano-, and alloyed compounds and materials, which have been successfully applied to numerous areas, such as the synthesis of pharmaceutical ingredients, catalysis, mineralogy, and even geology. Consequently, several insightful reviews and books that discuss the history and development of mechanochemistry and its application in these areas have been published[25–29]; further in-depth discussions on these established fields will not be included in this review.

Similar to the trajectory of Ar-BIAN complexes, mechanochemistry in the fields of synthesis and catalysis has predominantly been focused on transition-metal systems[30–32]. Nonetheless, the mechanochemical synthesis of main-group inorganic and organometallic compounds is a nascent and growing area. The first few examples of crystallographically characterized main-group complexes synthesized by mechanochemistry were a tris(allyl)aluminum complex[33] and a bis(n-propyltetramethylcyclopentadienyl)strontium complex[34] reported by Rheingold and Blair, respectively. In addition to organometallic small molecules, mechanochemical synthesis has also been successfully employed for the construction of non-carbon frameworks such as the tert-butyl-substituted adamantoid phosph(III)azane P$_4$(N$^t$Bu)$_6$, which was previously believed to be inaccessible due to the bulkiness of the tert-butyl group[35]. Other recent advances in the application of mechanochemical methods to main-group compounds have been summarized in recent reviews[27,28,36].

Specific to Ar-BIAN systems, mechanochemistry was utilized for the direct one-pot, two-step access to indium(III) complexes by milling acenaphthoquinone, the corresponding aniline, and InCl$_3$ in stainless-steel grinder jars for 30 min–2 h (Fig. 4a, b)[37]. Notably, this procedure bypassed the use of transition-metal halide-templating agents and could be generally used with anilines possessing both electron-donating and electron-withdrawing groups. Moreover, mechanochemistry facilitated the introduction of ester and carboxylate functionalities at the bay region of Ar-BIAN ligands (Fig. 4c)[38]. Based on the DFT calculations performed for five heteroleptic iridium complexes and supported by the Ar-BIAN ligands, the HOMO and LUMO of such complexes reside on the N,N-dimethylaniline and the acenaphthylene core of the Ar-BIAN ligands, respectively (Fig. 4d). Previously, Ar-BIAN compounds derived from 5-carboxymethylacenaphthoquinone via solution methods with ZnCl$_2$-templated condensation only proceeded in low yields owing to hydrolysis of the imines during removal of the metal template[11], which hindered purification of the ligand. On the

**Fig. 3 Synthetic routes to symmetrical and asymmetrical Ar-BIAN ligands. a** Typical solution-based synthetic route to Ar-BIAN compounds. **b** Synthesis of asymmetric Ar,Ar'-BIAN complexes via transimination from a symmetric Ar-BIAN Zn complex or ZnCl₂ condensation. **c** Synthesis of tridentate ligands comprising Ar-BIAN and pendant donors (O, P, and S).

contrary, an acid-catalyzed mechanochemical condensation successfully generated the desired ligand, highlighting how mechanochemical synthesis through ball milling is a powerful tool for the solid-state synthesis of Ar-BIAN ligands with facile purification steps[37].

**Ar-BIAN-supported catalysis.** In the arena of developing highly selective and efficient catalysts for industrially and pharmaceutically relevant chemical transformations, redox noninnocent ligands are especially popular since they offer additional charge-transfer capabilities[39]. However, the majority of catalysis supported by Ar-BIAN consists of transition-metal complexes, with some main-group counterparts being reported. Therefore, for completeness of the review and to best illustrate the potential of Ar-BIAN-supported catalytic processes, we briefly discuss the catalytic chemistry of the transition-metal Ar-BIAN complexes in addition to the *p*-block complexes.

Transition-metal complexes supported by redox noninnocent ligands have drawn considerable interest for the multielectron activation of small molecules and for catalytic reactions[39–41]. For example, Clark has recently reported the synthesis of [V(dmp-BIAN)₃](PF₆)] (dmp = 3,5-dimethylphenylimino), which was revealed to consist of two discrete redox isomers due to valence tautomerism[42]. Other mixed-ligand oxidovanadium(IV) complexes [VO(acac)(R-BIAN)][Cl] (R = H, CH₃, Cl) have also been

synthesized, and their catalytic activity toward olefin oxidation reactions was demonstrated[43]. Reaction of Dipp-BIAN with VCl₃ resulted in an oxidovanadium(IV) complex [[(Dipp-BIAN)VOCl₂]] with one unpaired electron. The electrochemical study on this complex revealed two quasi-reversible reductions at –0.32 and –1.05 V (Fig. 5a), followed by an irreversible reduction at –1.5 V (vs. Ag/AgCl). The first two reductions were assigned to the reduction of Dipp-BIAN, whereas the last reduction was presumably due to conversion of V(IV) into V(III). The EPR spectrum of [(Dipp-BIAN)VOCl₂] showed an eight-line isotropic signal characteristic of a $d^1$ electronic configuration, and the hyperfine interaction values fall in the typical range for oxidovanadium(IV) complexes (Fig. 5b). As shown from the magnetic susceptibility measurements, the $\mu_{eff}$ at 300 K is 1.67 $\mu_B$, which does not change with temperature. This consistent value of $\mu_{eff}$ together with a small Weiss constant suggested the absence of significant exchange interactions in the complex (Fig. 5c). The catalytic activity of this complex toward oxygenation of linear, cyclic, and branched alkanes by H₂O₂ was recently studied by Fomenko et al.[44]. Efficient oxidation was observed, especially in the presence of PCA (pyrazine-2-carboxylic acid), as revealed by the kinetic curves obtained for the formation of oxygenates from cyclohexane. A plausible mechanism was proposed by the authors, which involved the formation of an "activating complexation" system commonly observed for complexes of flavins and quinones. According to the DFT calculations, it was

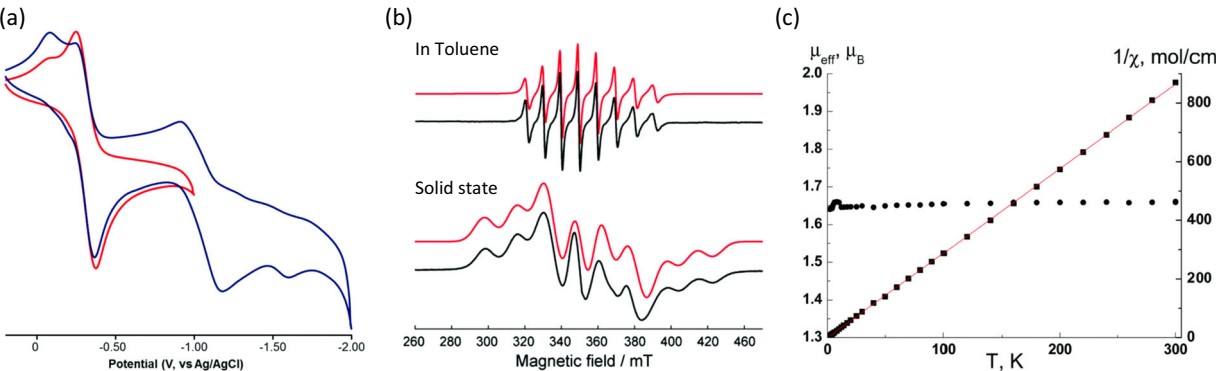

**Fig. 4 Mechanochemical synthesis of Ar-BIAN ligands and the characterization of their indium and iridium complexes. a** Mechanochemical synthesis of the Ar-BIAN ligands and indium(III) complexes. **b** Thermal ellipsoid plots (50%) of some of the Ar-BIAN indium(III) complexes, adapted from Wang et al.[37], under CC BY 3.0. **c** Mechanochemical synthesis of an ester-modified Ar-BIAN ligand. **d** Calculated frontier Kohn–Sham MOs of five iridium complexes, adapted from Hasan et al.[38], under CC BY. The symbol for mechanical milling was proposed by Hanusa et al.[98].

**Fig. 5 Electrochemical and spectroscopic characterization of the oxidovanadium(IV) complex [(Dipp-BIAN)VOCl₂].**

**Fig. 5 Electrochemical and spectroscopic characterization of the oxidovanadium(IV) complex [(Dipp-BIAN)VOCl$_2$]. a** Cyclic voltammograms of [(Dipp-BIAN)VOCl$_2$] in dichloromethane (DCM) at scan rates of 0.10 V s$^{-1}$. **b** X-band CW EPR spectra of [(Dipp-BIAN)VOCl$_2$] carried out at 298 K, with the simulations shown in red. **c** Temperature dependencies of $\mu_{eff}$ (solid circle) and $1/\chi$ (solid square) for [(Dipp-BIAN)VOCl$_2$]. Reproduced with permission from Fomenko et al.[44], Copyright 2018 by Royal Society of Chemistry.

proposed that the generation of HO• was associated with the redox-active nature of the BIAN ligand, and did not require a change in the metal oxidation state.

Another earth-abundant first-row transition metal that has been supported by sterically bulky BIAN ligands is Fe, as reported by Lei's group in 2017[15]. It had previously been shown that Ar-BIAN Fe(II) complexes catalyze the hydrosilylation of aldehydes and ketones at 70 °C under solvent-free condition[45]. To improve the efficiency of this catalytic reaction, increasing the steric bulk of the diimine substituents would be expected to help stabilize the Fe hydride intermediate formed. However, unlike palladium diimine complexes that showed significant variations in their catalytic activity with different sterically hindering groups[46], the differences in electronic and steric properties of the Ar-BIAN Fe(II) complexes did not appear to affect the catalytic reactivity for the hydrosilylation of various aldehydes and ketones[15]. In another example, the application of high-spin [(R-BIAN)FeCl$_2$] complexes for the hydrogenation of alkenes was reported by Wangelin in 2017[14]. An excess of a strong reductant, such as three equivalents of n-BuLi in toluene, was necessary to activate (Dipp-BIAN)FeCl$_2$ into [(Dipp-BIAN)Fe(PhMe)]$^-$Li$^+$ for the hydrogenation of olefins. Other examples of transition-metal complexes bearing redox noninnocent ligands that exhibit catalytic activity in hydrogenation reactions are acridine-containing pincer complexes. A recent report by Milstein described the synthesis of an acridine-based PNP pincer complex of Fe, which demonstrated catalytic activity in the selective hydrogenation of alkynes[47]. Along with the high yield, a high E selectivity has also been demonstrated for these pincer complexes, which was attributed to the fast isomerization of Z alkenes[48].

Moving down the period, it had been shown that Ar-BIAN Cu(I) complexes could be employed as light harvesters in both dye-sensitized solar cells and photoredox catalysis[16,49]. Owing to the remarkable π-accepting properties of the Ar-BIAN ligands, some of the Cu(I) complexes are panchromatic and absorb by metal-to-ligand charge transfer up to 1400 nm in the near-infrared region[49]. Later versions of these Cu(I) photosensitizers were reported by Soo's group, in which three Ar-BIAN-Cu(I) complexes were synthesized and their spectroelectrochemical properties were studied. Cyclic voltammetry (CV) measurements were performed to determine the feasibility of the Ar-BIAN-Cu(I) for mediating photoredox catalysis. During the cathodic scan, a quasi-reversible wave was observed at –0.99 V, suggesting possible complex regeneration after photocatalysis (Fig. 6a). To access the absorption and emission properties of the Cu(I) dyes, UV–vis and photoluminescence spectroscopic measurements were performed. As shown in Fig. 6b, the optical absorptions extended into the NIR region up to ~1000 nm, and a strong emission band was observed at 510 nm when the complexes were excited between 430 nm and 460 nm (Fig. 6c). Insights into their excited-state lifetimes were further obtained by time-correlated single-photon-counting spectroscopy (TCSPC), and recombination lifetimes of up to 11 ns were observed, which is longer than the rate of diffusion-controlled reactions and alludes to possible applications in photoredox catalysis (Fig. 6d)[16]. The Cu(I) complexes mediated the Karasch addition with C–C bond formation between styrene and CBr$_3$ radicals derived from CBr$_4$ in respectable yields via a radical chain propagation cycle[16].

In 2014, Lahiri reported the formation of five Ru-BIAN-based compounds [Ru(acac)$_2$(Ar-BIAN)] (Ar = Ph, 4-MeC$_6$H$_4$, 4-MeOC$_6$H$_4$, 4-ClC$_6$H$_4$, and 4-NO$_2$C$_6$H$_4$), which have been structurally, electrochemically, spectroscopically, and computationally characterized[50]. The strained carbon framework in the BIAN ligands offers a sensitive diagnosis for assessing the metal-to-ligand charge transfer in the resulting chelating compound. In a more recent article, the same group examined the synthesis and characterization of BIAN-based redox-active complexes [Ru(trpy)(Ar-BIAN)Cl][ClO$_4$] (trpy = 2,2′:6′,2″-terpyridine). These complexes were investigated for catalytic epoxidation reactions. It was found that electron-donating and -withdrawing groups on the para-position of the arylimino moiety of BIAN had little influence on the catalytic epoxidation process[51]. In addition, the catalytic activity of the newly synthesized Ru-BIAN complexes was explored in the oxidation of primary and secondary alcohols with H$_2$O$_2$ as the oxidant. These complexes demonstrated high selectivity toward the desired aldehydes and ketones since no overoxidized by-products were observed[51].

Examples of f-block complexes supported by Ar-BIAN ligands were first reported in 2007, and remained uncommon in the field of coordination chemistry[52–55]. In recent years, Niklas et al. described the synthesis of a modified BIAN ligand, namely Phen-BIAN, which integrates the acenaphthene backbone with a tetradentate mixed-donor O − N − N − O salophen-type binding motif[56]. Further treatment of the Phen-BIAN ligand with U(OAc)$_2$ resulted in the corresponding U(VI) complex in 84% yield. Structural characterization revealed that each metal center binds to one tetradentate ligand, and the complex adopts a dimeric aryloxide-bridged structure. Although catalytic studies have not been carried out, a wealth of redox activity and accessible oxidation states were suggested by the electrochemical studies, which portends future investigations of these complexes toward plausible catalytic reactions[56].

In contrast to the extensive work on transition-metal complexes, the diversity of catalytic systems mediated by main-group complexes has been comparatively small, and very few studies include Ar-BIAN ligands, despite their redox versatility. However, the number of examples of the main catalytic systems is rapidly increasing. In 2006, Harder introduced a calcium hydride complex that exhibited catalytic activity toward alkene hydrosilylation and hydrogenation[57]. More lately, examples of calcium benzyl and borate complexes have also been reported to be catalytically active in the hydroboration of 1,1-diphenylethylene (DPE) with catecholborane (HBcat)[58]. These constitute clear demonstrations that main-group organometallics are not limited to Lewis acidic and basic catalysis. In addition, Power in 2010 highlighted how heavier main-group elements can mimic transition metals in small-molecule (e.g., H$_2$ and NH$_3$) activation[59]. The new structural and bonding insights that heavier main-group elements can potentially exhibit multiple oxidation states and versatile coordination environments, make them attractive alternatives to conventional transition-metal-based catalysts.

With the emerging studies on many s- and p-block systems as effective catalysts in a multitude of reactions (e.g., cyclization, hydrogenation, and hydroaminaton), more attention has been turned to control enantioselectivity in addition to obtaining reasonable conversions. A recent review article by Wilkins and Melen described the competitive yields and enantioselectivities of catalytic C–C, C–H, C–N, C–O, and C–P bond-formation reactions shown by a new generation of catalysts composed of main-group elements[60]. These studies exploited the wide range of new opportunities offered by main-group chemistry in catalysis, an area previously dominated by heavy transition metals.

Nonetheless, we observed that the above-mentioned catalytic transformations involving noninnocent Ar-BIAN ligands within the main-group arena are still underexploited. In a recent example, the influence of Ar-BIAN's configurational rigidity and electronic lability on the chemical properties of the resulting complex can be illustrated through a digallane species [(Dipp-BIAN)Ga–Ga(Dipp-BIAN)], which is capable of activating alkyne triple bonds. Furthermore, the resulting complex is reactive

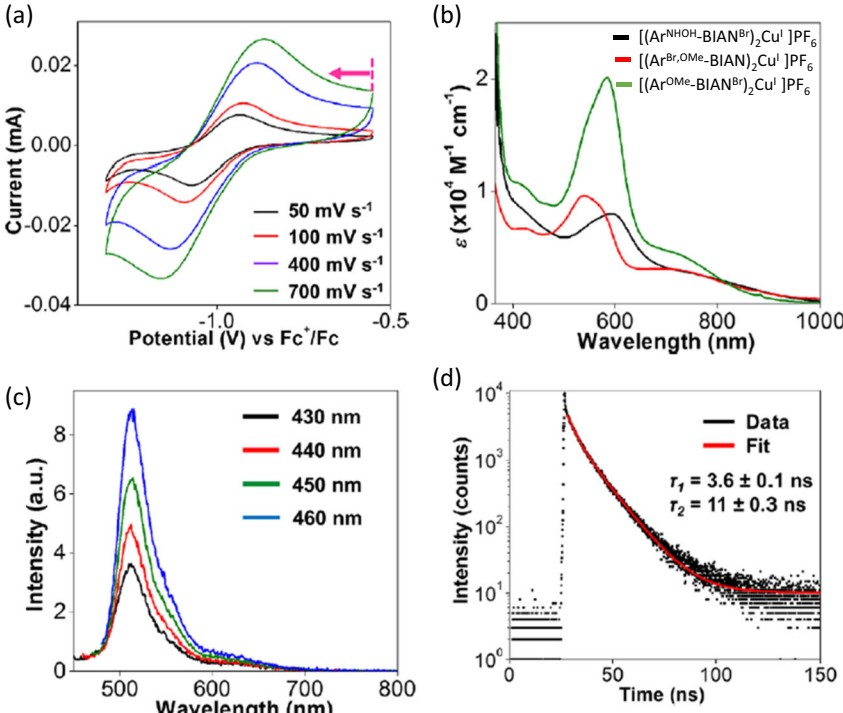

**Fig. 6 Electrochemical and spectroscopic properties of Ar-BIAN Cu$^I$ complexes. a** Cyclic voltammograms at different scan rates in the cathodic direction. **b** Electronic absorption spectra and **c** steady-state emission spectra with excitation from 430 to 460 nm. **d** Photoluminescence decay lifetimes after excitation at 466 nm. This figure was adapted with permission from Ng et al.[16], copyright 2018 by American Chemical Society.

toward electron-rich reagents, similar to olefins upon coordination to transition metals[61]. A plausible mechanism of the alkyne addition to [(Dipp-BIAN)Ga–Ga(Dipp-BIAN)] was proposed to involve a concerted interaction between the LUMO ($\pi^*$) of the alkyne with the HOMO ($\pi$) of the gallium complex. Alkyne molecules are added across the Ga–N–C fragment with regioselectivity controlled by electronic factors. To test for its catalytic reactivity toward new C–N bond-formation reactions, phenylacetylene was mixed with various anilines in the presence of [(Dipp-BIAN)Ga–Ga(Dipp-BIAN)]. It was found that this gallium complex serves well as a Markovnikov-selective catalyst for the hydroamination of PhC≡CH, with reaction rates comparable to other catalytic systems based on transition metals.

Overall, despite being clearly capable of controlling and fine-tuning catalytic processes, the incorporation of Ar-BIAN ligands in main-group catalysis remains a largely unexplored territory.

**Structural and electronic properties of *p*-block complexes of Ar-BIAN.** *Group 13 complexes*: The synthesis of 2-bromo-N,N′-bis(2′,6′-diisopropylphenyl)acenaphtho-1,3,2-diazaborole, appearing as burgundy red crystalline material, was achieved through the reaction of a diazaborolium salt with fivefold excess of 1% sodium amalgam in toluene[62]. The UV–vis spectrum of the resulting compound in CH$_2$Cl$_2$ showed a broad absorption band ranging from 450 to 600 nm, but exhibited no fluorescence. In contrast to 2-bromo-1,3,2-diazaboroles, which are prone to substitution of the Br by a wide range of nucleophiles like lithium aryls, the resulting diazaborole is reluctant to undergo such processes. DFT calculations suggest that the HOMO of this compound corresponds to the antibonding interaction of the acenaphthylene and the diazaborole framework, while the LUMO is mainly localized on the acenaphtho fragment. The electrostatic potential (ESP) map suggests a significant negative charge at the acenaphtho framework, which helps to explain the observed lack of reactivity toward even strong nucleophiles.

In 2002, Jenkins et al. reported the reaction of Dipp-BIAN with group 13 trihalides (BCl$_3$, AlCl$_3$, and GaCl$_3$) in a 1:2 ratio in dichloromethane (DCM). Three compounds with the general formula [(Dipp-BIAN)(ECl$_2$)][ECl$_4$] (E = B, Al, and Ga) were isolated (Fig. 7b)[63]. Later, the reactivity of gallium(I) and indium (I) halides toward Ar-BIAN ligands was examined by Jones's group in 2003[64]. The reaction of Dipp-BIAN with GaI in toluene led to disproportionation and the formation of a paramagnetic Ga (III) species [(Dipp-BIAN)•GaI$_2$] and presumably Ga(0), although the latter was not specified in the original reports[65,66]. In 2007, Fedushkin et al. reported the formation of Dipp-BIAN-chelated homobimetallic Ga–Ga and heterometallic Ga–Zn compounds[67]. [(Dipp-BIAN)Ga–Ga(Dipp-BIAN)], which is synthesized by the reaction of [(Dipp-BIAN)K$_3$] with GaCl$_3$ in Et$_2$O/THF at room temperature, is EPR silent and suggests that the two Ga atoms are in the +2 oxidation state, supported by the dianionic ene-diamide form of Dipp-BIAN. The related heterometallic [(Dipp-BIAN)Zn–Ga(Dipp-BIAN)] was obtained by a one-pot reaction between GaCl$_3$, [(Dipp-BIAN)K$_4$], and [(Dipp-BIAN)Zn(μ-I)$_2$Zn(Dipp-BIAN)], the last of which was synthesized from Dipp-BIAN and Zn metal in the presence of I$_2$[68]. The Zn–Ga bond distances in [(Dipp-BIAN)Zn–Ga(Dipp-BIAN)] fall between those observed in Zn–Zn homobimetallic complexes like [(Dipp-BIAN)Zn–Zn(Dipp-BIAN)], which is supported by two radical anionic Dipp-BIAN ligands[68,69], and [(Dipp-BIAN) Ga–Ga(Dipp-BIAN)], and is chelated by two dianionic Dipp-BIAN ligands. This suggests that the Dipp-BIANs in the heterometallic Ga–Zn complex are neither solely radical anions nor dianions, which has been complicated by the disorder between the Zn and Ga atoms in the crystallographic data[67]. The EPR spectrum of [(Dipp-BIAN)Zn–Ga(Dipp-BIAN)] also showed hyperfine coupling of the Zn atom to only two but not four N atoms.

The redox noninnocent behavior of BIAN ligands was established by Fedushkin's group when his group reported the

**Fig. 7 Examples of Ar-BIAN complexes with *p*-block elements. a** Behavior of the (Dipp-BIAN)Ga as a neutral σ donor and an anionic ligand toward Mo(0) and Mo(II), respectively. **b** Summary of Ar-BIAN complexes reported for the *p*-block elements.

formation of paramagnetic [(Dipp-BIAN)Ga–Mo(CO)$_5$] and diamagnetic [(Dipp-BIAN)Ga–MoCp(CO)$_3$] complexes. In [(Dipp-BIAN)Ga–Mo(CO)$_5$], the Ga carbenoid behaves as a neutral σ donor toward a Mo(0) center, while in [(Dipp-BIAN)Ga–MoCp(CO)$_3$], the Ga fragment acts as an anionic ligand toward a Mo(II) center (Fig. 7a). This adaptive behavior of the Ga carbenoids is enabled by the redox noninnocent Dipp-BIAN, which allows the reactivity at both Ga and the transition-metal center to be tuned[70]. The related complex [(Dipp-BIAN)Ga–Ga(Dipp-BIAN)] undergoes facile cycloaddition with alkynes, resulting in the formation of C–C and C–Ga bonds[71,72]. This unique reactivity was extended to the catalytic hydroamination of alkynes[61] with anilines with comparable activity to those of transition-metal-based systems[73,74] (vide supra). However, a similar reaction with dmp-BIAN as the ligand did not afford a compound with a metal–metal bond-like that formed with Dipp-BIAN. Instead, [(dmp-BIAN)$_2$Ga] with two ligands coordinated was generated, in which one dmp-BIAN is nominally dianionic, while the other one is a radical anion[75]. Addition of phenylacetylene to [(dmp-BIAN)$_2$Ga] was studied. The X-ray crystallographic data suggest that addition of the phenylacetylene occurred at the Ga–N–C group of the dianionic ligand. The regio- and stereoselectivity of addition is proposed to be controlled by both electronic and steric effects[75].

Although the complexation of Ar-BIAN ligands to heavier group 13 elements is less commonly reported compared with the lighter analogs, several crystallographically characterized examples of Ar-BIAN In(III) complexes have been recently synthesized[76]. Four different Ar-BIAN ligands were mixed with InCl$_3$ in THF to give the corresponding complexes. The ligands with *para*-substituents generally formed distorted octahedral complexes with coordinating solvents bound to In, while the ligands with *ortho*-substituents resulted in distorted square pyramidal In complexes. This behavior was attributed to the increased steric hindrance around In induced by the substituents at the *ortho* positions of the aniline.

Progressing to the next period, Mes-BIAN reacted with TlPF$_6$ to form [Tl(Mes-BIAN)$_2$][PF$_6$], which was the first example of a structurally authenticated Tl(I) complex of a neutral α-diimine ligand. It consists of a Tl atom linked to four imine N atoms from two Mes-BIAN ligands, forming a distorted square pyramidal geometry with the Tl atom at the vertex of the square pyramid[77].

*Group 14 complexes*: Fedushkin's group described the formation of three germylenes, [(Dipp-BIAN)Ge], [(dtb-BIAN)Ge] (dtb = 2,5-di-*tert*-butyl), and [(bph-BIAN)Ge] (bph = biphenyl) from metal exchange reactions of the respective magnesium Ar-BIAN complexes with GeCl$_2$[78]. A related Ge(II) compound [(Dipp-BIAN)GeCl] bearing a radical anionic Ar-BIAN ligand can be prepared either by reacting the free Dipp-BIAN ligand with two equivalents of GeCl$_2$ or by metathesis of [(Dipp-BIAN)Na] with GeCl$_2$[79]. Similar to the Dipp-BIAN ligand, dtb-BIAN also reacts with GeCl$_2$ to form [(dtb-BIAN)GeCl], in which the

ligand forms a radical anion[80]. The EPR spectra for both complexes revealed coupling between the unpaired electron and the Ge, Cl, and N nuclei. However, despite their structural similarity, the $^{73}$Ge, $^{35}$Cl, and $^{37}$Cl hyperfine coupling constants in [(Dipp-BIAN)GeCl] and [(dtb-BIAN)GeCl] are significantly different. This disparity was attributed to the difference in contributions of the germanium $s$ and $p$ orbitals to the Ge–Cl bond. The larger the contribution of the $s$ orbital to the Ge–Cl bond, the larger are the Ge and Cl hyperfine coupling constants.

With the aim of synthesizing tin analogs of the germylenes, metathesis of $SnCl_2$ with the disodium salt of Dipp-BIAN ligand [(dipp-BIAN)$Na_2$] was performed[80]. However, this reaction only led to the formation of elemental tin and free Dipp-BIAN ligand. On the other hand, reaction of $SnCl_2$ with the Mes-BIAN ligand resulted in disproportionation of the metal center, forming the Sn (IV) compound [(Mes-BIAN)$SnCl_4$] and tin metal[81]. Compound [(Mes-BIAN)$SnCl_4$] consists of a neutral Mes-BIAN ligand coordinating to the $SnCl_4$ unit via formally dative bonds, and features the same coordination environment as [(Dipp-BIAN)$SnCl_4$] formed by treating Dipp-BIAN directly with $SnCl_4$[77]. In contrast to the Mes-BIAN ligand, the reaction of $SnCl_2$ with the dtb-BIAN ligand did not lead to disproportionation of the metal center, but gave the corresponding Sn(II) complex [(dtb-BIAN) $SnCl_2$] instead[80]. X-ray crystallographic analysis on [(dtb-BIAN) $SnCl_2$] revealed that the $SnCl_2$ is coordinated to only one N atom from the dtb-BIAN ligand (Fig. 7b). However, its $^1H$ NMR spectrum in $C_6D_6$ suggested the presence of a mirror plane bisecting the N–C–C–N framework in solution since the *tert*-butyl groups on each diimine moiety appeared to be magnetically equivalent. This suggests that the $SnCl_2$ fragment exchanges rapidly between the two diimine N atoms in solution on the timescale of the experiment.

*Group 15 and 16 complexes*: Reeske et al. reported the first application of Ar-BIAN ligands in the context of group 15 chemistry, which focused on the reaction of $PCl_3$ or $AsCl_3$ with Dipp-BIAN[82] (Fig. 7b). It was demonstrated that the reduction of $PCl_3$ with $SnCl_2$ in the presence of bis(phosphine) ligands resulted in the formation of cyclic triphosphenium ions[83]. In this reaction, an initial formation of "PCl" and $SnCl_4$ occurred via reduction of $PCl_3$ with $SnCl_2$. The "PCl" species was then coordinated to the bis(phosphine) ligand concomitant with or prior to a halide abstraction by $SnCl_4$. Inspired by these findings, Reeske et al. investigated the result of trapping the "PCl" species with ligands other than phosphines, e.g., the Ar-BIAN ligands[82]. Treatment of Dipp-BIAN with one equivalent of $PCl_3$ and $SnCl_2$ afforded the phosphenium salt [(Dipp-BIAN)P][$SnCl_5$·THF]. The related arsenium salt [(Dipp-BIAN)As][$SnCl_5$·THF] could also be obtained with $AsCl_3$ using the same protocol. X-ray structural analysis of these complexes revealed a $PN_2C_2$ ring with the C–C and C–N bond distances corresponding to a C=C double bond and C–N single bond, respectively. Hence, the Dipp-BIAN ligand appeared to be reduced by two electrons, while the oxidation state of the P or As center remained as +3. The authors also studied the reaction of Dipp-BIAN with $PI_3$ without $SnCl_2$ as the reducing agent. This reaction resulted in quantitative formation of the phosphenium triiodide salt [(Dipp-BIAN)P][$I_3$], which also possessed a phosphenium cation as indicated by the $^{31}$P NMR data.

The antimony (Sb) and bismuth (Bi) complexes supported by Ar-BIAN ligands were obtained from the reaction of Ar-BIAN directly with the halide salts of the corresponding pnictogen. Treatment of Dipp-BIAN with an equimolar amount of $SbCl_3$ in dichloromethane furnished the expected Sb(III) complex [(Dipp-BIAN)$SbCl_3$][77]. Based on the X-ray crystallographic analysis, the Sb(III) metal center exhibits a distorted square-pyramidal geometry, in which the vacant equatorial position is occupied by the lone pair of the nonbonding electrons on Sb. The two N–Sb bond distances are very different (2.617 Å and 2.846 Å), and both are significantly longer than those reported for a related Sb(III) diimine complex [$^t$Bu-DAB($SbCl_3$)$_2$][84]. This suggests that despite its facile formation, the N–Sb bonds in [(Dipp-BIAN) $SbCl_3$] are relatively weak as compared with those of other Sb(III) complexes supported by diimine ligands.

The application of Dipp-BIAN to support cations involving fluoro-, azido-, and cyano-antimony(III) moieties was later reported by Ferguson in 2017[85]. Two fluoro-antimony complexes [SbF(Dipp-BIAN)][OTf]$_2$ and [$SbF_2$(Dipp-BIAN)][OTf] (OTf = trifluoromethanesulfonate) were successfully isolated and characterized[85]. They reported the reactivity of the cationic coordination complexes of Sb with Dipp-BIAN ligand. Reactions of SbF(OTf)$_2$ and $SbF_2$(OTf) with Dipp-BIAN in dichloromethane resulted in the formation of Sb(III) complexes [SbF (Dipp-BIAN)](OTf)$_2$ and [$SbF_2$(Dipp-BIAN)](OTf), respectively. Further treatment of the former with trimethylsilyl cyanide (TMS-CN) and azidotrimethylsilane (TMS-$N_3$) gave the corresponding cyano-Sb(III) complex [Sb(CN)(Dipp-BIAN)] (OTf)$_2$ and azido-Sb(III) complex [$SbN_3$(Dipp-BIAN)](OTf)$_2$, respectively. These reactions represent examples of using Ar-BIAN ligands to support cations involving fluoro-, azido-, and cyano-antimony moieties[85]. Similar to the preparation of Sb(III) Ar-BIAN complexes, the Bi(III) Ar-BIAN complexes [(Dipp-BIAN)$BiCl_3$] and [(Mes-BIAN)$BiCl_3$] were synthesized by reaction of the corresponding Ar-BIAN ligand with $BiCl_3$[77]. The solid-state structure of [(Mes-BIAN)$BiCl_3$] shows a $\mu$-Cl-bridged dimer, where each of the two Bi atoms are bridged by $Cl^-$ anions.

Besides the heavier pnictogens, there has also been some exploratory work on the chemistry of group 16 BIAN complexes. The reaction of $TeI_4$ with Mes-BIAN and Dipp-BIAN resulted in two-electron reduction of the metal center and formation of (Mes-BIAN)$TeI_2$ and (Dipp-BIAN)$TeI_2$, respectively[86]. The bond distances obtained from the X-ray crystallographic analysis each revealed a neutral BIAN ligand coordinating to a Te(II) center, forming a square planar geometry around Te. In contrast, spectroscopic evidence suggested that treatment of $TeCl_4$ with Dipp-BIAN led to the formation of a 1:1 complex without reduction of the Te(IV).

**Physicochemical properties and application of noninnocent ligands in catalysis**. To date, the literature on main-group Ar-BIAN complexes has focused on synthetic and structural studies with no major efforts devoted to the investigation of their physicochemical properties. In-depth studies were first described in 2016, where the spectroscopic and electrochemical properties of two Ar-BIAN indium(III) complexes were presented[37]. In general, the extinction coefficient is lower for complexes formed with an electron-withdrawing group (e.g., *p*-BrAr-BIAN-$InCl_3$) than that with an electron-donating group (e.g., *p*-MeOAr-BIAN-$InCl_3$) (Fig. 8a). The same trend was observed when comparing the ligands *p*-BrAr-BIAN and *p*-MeOAr-BIAN, which was consistent with similar observations reported by Hasan et al.[20] The higher-energy bands below 350 nm and the lower-energy bands above 400 nm may be assigned to the π–π* ligand-centered transitions from within both diimine and acenaphthene, and between diimine and acenaphthene, respectively, similar to the assignments reported in the literature for related Ar-BIAN systems[87,88]. These UV–vis spectral band assignments were also supported by the electron-density distribution, which showed that the HOMO mainly consists of the arylimino fragments from Ar-BIAN, whereas the LUMO is delocalized over the acenaphthene moiety (Fig. 8b).

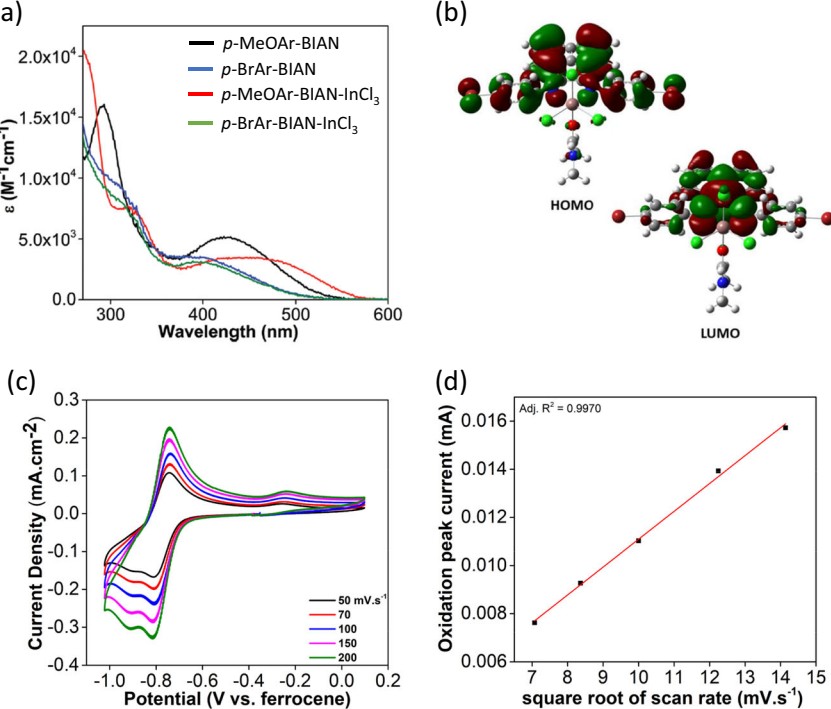

**Fig. 8 Spectroscopic, structural, and electrochemical characterization of Ar-BIAN indium(III) complexes. a** UV–vis spectra for *p*-MeOAr-BIAN-InCl₃ and *p*-BrAr-BIAN-InCl₃ in different solvents. **b** Electron-density distributions of the molecular orbitals involved in the UV–Vis spectra. **c** The first and second reduction waves of *p*-MeOAr-BIAN-InCl₃ at different scan rates. **d** A plot of the peak current for *p*-MeOAr-BIAN-InCl₃ at $E_p^{oxi} = +0.74$ V versus the square root of the scan rate. Adapted from Wang et al.[76], under CC BY 3.0.

The electrochemical behavior of the indium(III) Ar-BIAN complexes was also studied by cyclic voltammetry. During the cathodic scan of *p*-MeOAr-BIAN-InCl₃, a total of six reduction waves were observed. The first reduction at $E_p^{red} = -0.80$ V is chemically reversible because $\Delta E_p = 60$ mV, and the ratio of the cathodic and anodic peak currents stays around one, regardless of the scan rate (Fig. 8c). This was further confirmed by a linear fit of the oxidation peak current (where $E_p^{oxi} = +0.74$ V) plotted against the square root of the scan rate (Fig. 8d). In contrast, no chemically reversible processes were identified in the case of *p*-BrAr-BIAN-InCl₃, with only two quasi-reversible reduction waves observed between $E_p^{red} = -1.0$ V and $E_p^{red} = -2.2$ V. It was proposed that these two reduction waves arise from the reduction of the In(III) metal center in *p*-BrAr-BIAN-InCl₃[37]. When the metal is reduced, one of the Cl⁻ dissociates, which will undoubtedly result in structural changes to the molecule. This in turn leads to a reoxidation process at a different potential. Subsequent reduction processes observed in *p*-BrAr-BIAN-InCl₃ are probably due to ligand reduction. The fact that reduction waves attributed to the reduction of In(III) metal center appeared to be electrochemically reversible for *p*-MeOAr-BIAN-InCl₃, but not in the case of *p*-BrAr-BIAN-InCl₃, suggested an increased stability of the putative [(*p*-MeOAr-BIAN)₂InCl₃]⁻ as compared with [(*p*-(BrAr-BIAN)InCl₃]⁻.

Similar studies on the absorption and electrochemical behavior have been expanded to indium(III) complexes with a larger variety of Ar-BIAN ligands in a later report[76]. It was concluded that all the indium(III) Ar-BIAN complexes discussed absorb light in the UV and visible regions of the electromagnetic spectrum, with complexes bearing strongly electron-donating substituents absorbing further to the red region due to the reduced HOMO–LUMO gap of the complexes. Also, chemically reversible reduction waves were only observed for complexes with electron-donating substituents on the arylimino fragment,

suggesting an increased stability of the putative reduced species in these complexes as compared with those having electron-withdrawing substituents.

Beyond fundamental studies and characterization, the application of noninnocent ligands in catalysis can be classified into four different strategies, as summarized by de Bruin[89]. The first involves oxidation and reduction of the ligand to accommodate the electronic changes to the metal center. In the second approach, the ligand serves as an electron reservoir, which allows for multiple electron transformations in metal complexes where the metal itself is reluctant to undergo redox changes. The third approach involves the generation of ligand radicals that participate in bond formation during catalysis. The last strategy requires the activation of the substrate in cases where the substrate binds as a redox noninnocent ligand as well[89]. The first approach can be illustrated by the oxidation of H₂ using Cp*Ir (ᵗBAᶠPh), where H₂ᵗBAᶠPh is 2-(2-trifluoromethyl)anilino-4,6-di-*tert*-butylphenol. This cationic complex contains a one-electron oxidized ligand radical cation, which results in increased Lewis acidity of the metal center[90]. The second approach is more often observed in homogeneous catalysis where multiple electron transfers occur between the complex and the substrate. While this process seems trivial for late- transition metals, it is more difficult to achieve with early transition metals due to the lack of easily accessible, contiguous oxidation states. This highlights how redox noninnocent ligands can serve as electron reservoirs. In these first two approaches, the catalytic reactivity occurs at the metal center, meaning that each redox noninnocent ligand plays the role of a "spectator". In contrast, the ligands play more "active" roles in the last two approaches, where bond formation and breakage with the substrate are expected.

In this context, one of the well-known applications exhibited by Ar-BIAN complexes is in ring-opening polymerization (ROP). For instance, [(Ar-BIAN)ZnMe(THF)]⁺ demonstrated high ROP

activity for ε-caprolactone with excellent molecular weight control[91]. Among main-group complexes, promising ROP of cyclic esters was also shown by In(III) complexes, as illustrated recently by a cationic salen-type In complex[92]. Cationic Ga and Al complexes supported by Dipp-BIAN were lately reported by Fedushkin to show similar catalytic activity[93]. An electron-deficient gallylene $[(Dipp-BIAN)Ga]B(C_6F_5)_3$ complex was formed through the reaction of $[(Dipp-BIAN)Ga−Ga(Dipp-BIAN)]$ with $B(C_6F_5)_3$, and it was shown to serve as an initiator for the ROP of ε-caprolactone[59,93].

**Outlook**. Transition-metal catalysis has been the workhorse in organic and organometallic chemistry, owing to the accessibility of multiple oxidation states and coordination environments of the complexes. Lately, however, more insights into the bonding, structural, and electronic properties of main-group complexes have been obtained, leading to greater efforts being devoted to the study of catalytic reactions facilitated by main-group complexes[59]. For example, Power highlighted the appearance of nonbonded electron density and concomitant geometrical distortions in heavy main-group metal complexes, which led to the previously unknown small-molecule activation reactions of $H_2$, $NH_3$, CO, and $C_2H_4$[59]. Other redox noninnocent ligands, such as N-heterocyclic carbenes, have further enriched the landscape of main-group coordination chemistry and catalytic applications[12,14,94].

Very recently, Bertrand's group reported the isolation of monosubstituted carbenes, which are the first examples of monosubstituted carbenes isolable at room temperature, and link the rich chemistry of disubstituted triplet carbenes with the transient parent carbene[95]. In addition, six-membered cyclic (alkyl)(amino)carbenes have also been reported and demonstrated superior reactivity in the arylation of ketones with aryl chlorides relative to the traditional five-membered N-heterocyclic carbenes[96]. Jones et al. had also reviewed the redox noninnocence of heavy main-group antimony complexes supported by ambiphilic ligands[97], with the intention of applying them toward transition-metal–halide-bond activation reactions to expose active sites for further catalysis. Thus, the synergy of main-group elements with redox noninnocent ligands remains a very active area of exploration.

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

## Acknowledgements

F.G. acknowledges A*STAR (AME IRG A1783c0003) and a NTU start-up grant (M4080552) for financial support. H.S.S. is supported by Singapore Ministry of Education Academic Research Fund Tier 1 grants RG 111/18 and RT 05/19, and also Agency for Science, Technology and Research (A*STAR) AME IRG grants A1783c0002, A1783c0003, and A1783c0007.

## Author contributions

J.W., H.S.S., and F.G. selected and discussed the pertinent literature examples, composed the narrative, and reviewed the paper.

## Competing interests

The authors declare no competing interests.
