## [Peer Review File · Communications Chemistry]

REVIEWERS' COMMENTS:

Reviewer #1 (Remarks to the Author):

This review is entitled: "Latest breakthroughs in p-block complexes supported by bis(arylimino)acenaphthene ligands". While the chemistry described herein is interesting, I have a couple of issues in general with this review.

1. From the title one would think the chemistry detailed would be focused on main group systems and yet the authors present a great deal of information on transition metal chemistry as well as a section detailing "Transition metal complexes of Ar-BIAN for catalysis". This seems odd and out of place for the subject.
2. I am also curious about the selection of the subject. This ligand is interesting but it is just one of a class of poly-aromatic diamines. It seems the authors have narrowed the scope so finely that it might be of interest to a rather thin section of the main group inorganic community rather drawing interest from a broader cross section of chemists.

Reviewer #2 (Remarks to the Author):

Wang, Soo and García provide a concise review that covers the recent advances made in the field concerning the chemistry of Ar-BIAN main group complexes. It gives the reader an insight into what progress has been made in their preparation that includes approaches involving the current hot-topic area of solid-state synthesis. Along with this, the authors give an account of the latest developments in the structural characterisation, and the optical and electrochemical studies of Ar-BIAN main group complexes. Overall, this is a well-written and timely review of the field which will be of interest to the general readership of Commun. Chem. and publication is recommended once the following corrections/revisions have been made: (1) the opening gambit of the abstract has a somewhat negative slant with the use of discouraging adverbs and adjectives such as however and even. It is the view of this reviewer that this should be avoided and to simply state what had been achieved prior to the current period of this review; (2) throughout the manuscript, a number of the sentences are somewhat long. For example, the sentence on page 4 beginning with: "As illustrated in Figure 2b,....." might read better as: "As illustrated in Figure 2b, reduction of Ar-BIAN by a one electron process first produces a radical anion that is delocalized over the NCCN fragment. This is followed by a second electron reduction, which results in the formation of a dianionic ene-diamide."; (3) "p-block" in the sub-titles on pages 14 and 21 should read as "p-Block" for consistency; (4) Refs. 4-8, 10, 12, 14, 41, 42, 44, 60, 70, 71, and 81 containing formatting errors and need to be corrected.

REVIEWERS' COMMENTS:

Reviewer #1 (Remarks to the Author):

This review is entitled: "Latest breakthroughs in p-block complexes supported by bis(arylimino)acenaphthene ligands". While the chemistry described herein is interesting, I have a couple of issues in general with this review.

1. From the title one would think the chemistry detailed would be focused on main group systems and yet the authors present a great deal of information on transition metal chemistry as well as a section detailing "Transition metal complexes of Ar-BIAN for catalysis". This seems odd and out of place for the subject.

We would like to thank Reviewer 1 for this feedback. Following Reviewer 1's advice, the title for the section "Transition metal complexes of Ar-BIAN for catalysis" has been changed to "Ar-BIAN supported catalysis". The revised section has been expanded to start with a discussion about catalysis associated with transition metal Ar-BIAN complexes to best illustrate the capabilities of this type of ligands, followed by a dedicated discussion on main group catalysis to conclude.

2. I am also curious about the selection of the subject. This ligand is interesting but it is just one of a class of poly-aromatic diamines. It seems the authors have narrowed the scope so finely that it might be of interest to a rather thin section of the main group inorganic community rather drawing interest from a broader cross section of chemists.

The class of poly-aromatic diamines is broad and covers both redox innocent and non-innocent ligands. A number of review papers have been published discussing the

coordination chemistry with these diamines. However, there are no recent reviews that focus on Ar-BIAN ligands. The aim of this review is not to provide a comprehensive discussion about the entire family of diamines as a whole since that would not be appropriate for this journal's format. Instead, we intended to provide a selected, but still, a general overview of main group Ar-BIAN complexes, focusing primarily on the structural, physicochemical, and prospective use in main group catalytic processes. We aim to provide a concise description to the community working on redox non-innocent diimine ligands, which can inspire more researchers to consider incorporating Ar-BIAN ligands in their catalyst design.

Reviewer #2 (Remarks to the Author):

Wang, Soo and García provide a concise review that covers the recent advances made in the field concerning the chemistry of Ar-BIAN main group complexes. It gives the reader an insight into what progress has been made in their preparation that includes approaches involving the current hot-topic area of solid-state synthesis. Along with this, the authors give an account of the latest developments in the structural characterization, and the optical and electrochemical studies of Ar-BIAN main group complexes. Overall, this is a well-written and timely review of the field which will be of interest to the general readership of Commun. Chem. and publication is recommended once the following corrections/revisions have been made:

- (1) the opening gambit of the abstract has a somewhat negative slant with the use of discouraging adverbs and adjectives such as however and even. It is the view of this reviewer that this should be avoided and to simply state what had been achieved prior to then current period of this review;

We thank Reviewer 2 for investing time and effort to review our manuscript. Following Reviewer 2's advice, we have removed the words with negative connotations (e.g. however, although) from the preface.

- (2) throughout the manuscript, a number of the sentences are somewhat long. For example, the sentence on page 4 beginning with: "As illustrated in Figure 2b,....." might read better as: "As illustrated in Figure 2b, reduction of Ar-BIAN by a one electron process first produces a radical anion that is delocalized over the NCCN fragment. This is followed by a second electron reduction, which results in the formation of a dianionic ene-diamide.";

This sentence and other lengthy ones have been amended to be more concise following Reviewer 2's recommendations.

- (3) "p-block" in the sub-titles on pages 14 and 21 should read as "p-Block" for consistency;

The titles have been corrected and they are now consistent.

- (3) Refs. 4–8, 10, 12, 14, 41, 42, 44, 60, 70, 71, and 81 containing formatting errors and

need to be corrected.

All the reference have been checked and corrected where required.

We hope that these amends address the reviewers' concerns. The authors appreciate your time for considering our manuscript and look forward to your positive response.

Looking forward to hearing back from you.

Yours Sincerely,

Felipe García